# Combatting Sugar Beet Root Rot: *Streptomyces* Strains’ Efficacy against *Fusarium oxysporum*

**DOI:** 10.3390/plants13020311

**Published:** 2024-01-20

**Authors:** Walaa R. Abdelghany, Abeer S. Yassin, Farrag F. B. Abu-Ellail, Areej A. Al-Khalaf, Reda I. Omara, Wael N. Hozzein

**Affiliations:** 1Plant Pathology Research Institute, Agricultural Research Center, Giza 12619, Egypt; 2Sugar Crops Research Institute, Agricultural Research Center, Giza 12619, Egypt; 3Biology Department, College of Science, Princess Nourah bint Abdulrahman University, Riyadh 11671, Saudi Arabia; 4Botany and Microbiology Department, Faculty of Science, Beni-Suef University, Beni-Suef 62511, Egypt

**Keywords:** sugar beet, biological control, actinobacteria, *Streptomyces*, *Fusarium oxysporum*, root rot

## Abstract

Sugar beet root rot disease triggered by *Fusarium oxysporum* f. sp. *radicis-betae* is a destructive disease and dramatically affects the production and quality of the sugar beet industry. Employing beneficial microorganisms as a biocontrol strategy represents an eco-friendly and sustainable approach to combat various plant diseases. The distinct aspect of this study was to assess the antifungal and plant growth-promoting capabilities of recently isolated *Streptomyces* to treat sugar beet plants against infection with the phytopathogen *F. oxysporum*. Thirty-seven actinobacterial isolates were recovered from the rhizosphere of healthy sugar beet plants and screened for their potential to antagonize *F. oxysporum* in vitro. Two isolates SB3-15 and SB2-23 that displayed higher antagonistic effects were morphologically and molecularly identified as *Streptomyces* spp. Seed treatment with the fermentation broth of the selected *Streptomyces* strains SB3-15 and SB2-23 significantly reduced disease severity compared to the infected control in a greenhouse experiment. *Streptomyces* SB2-23 exhibited the highest protective activity with high efficacy ranging from 91.06 to 94.77% compared to chemical fungicide (86.44 to 92.36%). Furthermore, strain SB2-23 significantly increased plant weight, root weight, root length, and diameter. Likewise, it improves sucrose percentage and juice purity. As a consequence, the strain SB2-23’s intriguing biocontrol capability and sugar beet root growth stimulation present promising prospects for its utilization in both plant protection and enhancement strategies.

## 1. Introduction

Sugar beet (*Beta vulgaris* var. *saccharifera*, L.) serves as an essential and indispensable sugar supply for the natural sweetener industry. Due to its lower water consumption compared to sugar cane, Egypt is focusing on expanding its domestic sugar production by fostering sugar beet cultivation to bridge the sugar deficit. However, the sugar crop faces recurring challenges from various biotic pathogens including fungi, bacteria, viruses, and nematodes, leading to substantial declines in production [1,2,3,4]. *Fusarium* species are among the most dangerous fungi that cause crop root infections, which diminish root production as well as sugar percentage and juice purity [5]. These pathogens cause different destructive diseases on the crop including Fusarium yellows triggered by *F. oxysporum* f. sp. *betae* [6], Fusarium root rot triggered by *F. oxysporum* f. sp. *radicis-betae* [7,8], Fusarium stalk rot induced by *F. solani* [9], and the rotting of stored beet root caused by *F. culmorum*, *F. cerealis*, *F. redolens*, and *F. graminearum* [10]. 

Fusarium root rot has become increasingly common, particularly across numerous sugar beet growing areas, potentially as a result of the increased planting of susceptible varieties. The disease is distinguished by leaf withering, interveinal chlorosis, and discoloration of the root’s vascular tissues, often leading to the death of the plant [11]. Furthermore, the pathogen produces an extensive amount of macroconidia, microconidia, and chlamydospores that may persist in the soil for more than 10 years and infect over 80% of sugar beet cultivars [12]. Given the toxicity potential, the highly diverse range of Fusarium species associated with sugar beet, and economic losses, crop management against these pathogens is critical. Although chemical fungicides are commonly employed to manage many plant diseases, their abuse and indiscriminate usage have resulted in severe impacts on human, animal, and environmental health. As a result, its applicability should be restricted [3]. Furthermore, climate change and the emergence of fungicide resistance are also reducing the efficacy of synthetic chemicals. Biological management has evolved as a viable, effective, and safe alternative technique for managing fungal phytopathogens in recent decades [13]. A novel approach in disease management and plant growth enhancement entails utilizing bacterial and fungal bioagents instead of relying on synthetic chemical inputs [14,15,16]. Isolating microorganisms, particularly actinobacteria, stands out due to their capability to produce secondary metabolites like antibiotics and extracellular enzymes. This method is highly efficient and effective in the discovery of new bioactive metabolites [17]. Almost 80% of antibiotics are derived from actinobacteria, mostly from the genera *Streptomyces* and *Micromonospora* [18,19]. *Streptomyces* alone is accountable for roughly 75% of most bioactive compounds, which encompasses antibiotics [20,21]. The capacity of this genus to synthesize antibiotics and various other bioactive components provides an evolutionary advantage, allowing it to adapt to various and variable stressful conditions. Furthermore, because of the complexity of their metabolism, they have been able to colonize many ecosystems and employ a wide array of carbon and nitrogen sources.

In the same regard, Actinobacteria, mainly *Streptomyces*, have emerged as crucial contributors to play a significant role in controlling soil-borne plant pathogens in rhizosphere soil. They achieve this by generating enzymes that degrade fungal cell walls as well as producing antifungal compounds [13,22]. Additionally, actinobacteria may safeguard plant roots from infections and stimulate plant development through the release of plant growth-promoting compounds, minerals, and nutrients, or boosting the proliferation of beneficial microbes [23,24]. Actinobacteria’ modes of action for plant root protection have been reported to include antibiosis, parasitism, the production of extracellular hydrolytic enzymes, competition for iron, and the induction of systemic resistance in the host plant [21,25]. Continued research endeavors are vital to bolster disease resistance, explore novel biological control agents, and establish sustainable management practices. Employing this multifaceted approach is essential to safeguarding sugar beet crops and ensuring stable production despite the challenges posed by *Fusarium*-related diseases. This study aimed to isolate *Streptomyces* strains from the rhizosphere of healthy sugar beet plants and assess their potential as biocontrol agents against the fungal phytopathogen *F. oxysporum*, aiming to understand their interaction and potential positive impacts on sugar beet. These findings are valuable in the quest for novel natural bioagents possessing potent antifungal activities for the sugar beet industry.

## 2. Material and Methods

### 2.1. Sampling and Isolation of Actinobacteria

Sugar beet plants were collected in sterile bags from different governorates of Egypt, Beni-Suef (28°91′ N; 30°95′ E), Giza (30°02′ N; 31°20′ E), Fayoum (29°42′ N; 31°03′ E), and Kafrelsheikh (31°06′ N; 30°84′ E) during the 2020/2021 growing season. Five samples were collected from each location. Plant roots were shaken gently to eliminate loosely adhering soils. The soil samples were air-dried by spreading samples over sheets of paper at room temperature for 15 days to reduce the number of vegetative bacteria. Soil samples from each site were then mixed and subsequently used for the isolation.

The selective isolation of actinobacteria from the rhizosphere of healthy sugar beet plants was performed via soil dilution plate technique method on starch casein medium. The composition of this medium consisted of soluble starch (10 g), KNO_3_ (2 g), casein (0.3 g), K_2_HPO_4_ (2 g), MgSO_4_.7H_2_O (0.05 g), CaCO_3_ (0.02 g), FeSO_4_.7H_2_O (0.01 g), and agar (15 g). 

The process began by serially diluting air-dried soil samples by aseptically adding 1 g of soil to 9 mL of sterilized distilled water (10^−1^), the mixture was shaken vigorously using a vortex, followed by further tenfold dilutions up to 10^−4^. Each soil dilution was distributed by dropping 100 µL of it over the surface of the isolation plates and spreading it with a sterile glass spreader under aseptic conditions. Five replicate plates were made for each dilution. The plates were then incubated for 15 days at 30 °C, which is an optimal temperature for actinobacterial growth. 

Colonies of actinobacteria were selected according to their morphological characteristics and were purified several times by streaking in the same isolation medium and subsequently used for further studies. For long-term preservation, cultures were kept at −80 °C in 15% glycerol. Glycerol acted as a cryoprotectant, preventing ice crystal formation during freezing and safeguarding the actinobacterial cells from damage.

### 2.2. Source of Fungal Pathogen

The pathogenic *Fusarium oxysporum* f. sp. *radicis-betae* F186, previously isolated from diseased sugar beet plants exhibiting typical Fusarium root rot symptoms [26] was generously provided by the culture collection of the Maize and Sugar Crops Department at the Plant Pathology Research Institute, Agricultural Research Center in Giza, Egypt. For subsequent research purposes, it was cultured on potato dextrose agar medium (PDA).

### 2.3. In Vitro Screening of Antagonistic Activity

The antagonistic effect of isolated actinobacteria against the fungal pathogen was performed through an in vitro plate confrontation method on potato dextrose agar (PDA) medium. Fungal cylinders of well-grown and sporulated 7 days old culture agar discs (8 mm diameter) were centrally placed on PDA plates. Meanwhile, a loopful of each isolated actinobacterium at 7 days old was streaked along the periphery of the plate, maintaining a distance of 3 cm from the fungal disc. Control plates contained fungal discs without actinobacterial application. All plates were left to incubate at 28 °C for 5 days or till the fungal pathogen populated the entire surface of the control plates. Fungal growth cessation or the discernible zones of inhibition were considered a criterion for positive effect. Additionally, the average colony diameter of phytopathogenic fungus was assessed in both the treatment and control groups. The fungal radial growth inhibition rates were estimated using the formula $Inhibition\;rate=\frac{C-T}{C}\times 100$, where *C* and *T* represented the mean colony diameters (cm) of phytopathogenic fungi in the control and treatment, respectively. 

### 2.4. Identification of the Most Potent Actinobacteria

The cultural characteristics of the selected potent actinobacteria, which showed a high ability to produce antimicrobial metabolites were investigated on the International *Streptomyces* Project ISP media. Observations included growth patterns, the coloration of mature sporulating aerial surfaces, substrate mycelium color as viewed from the opposite side, and diffusible soluble pigments. Cultures aged 7 days were assessed on ISP1, ISP2, ISP4, ISP6, and PDA media to ascertain these characteristics. The determination of colors was conducted through comparison with chips available from the ISCC-NBS color charts (Standard Samples No. 2106).

The amylolytic and protease activity of the potent actinobacteria were tested by inoculating them on starch agar and skim milk agar medium, respectively [27]. Catalase and gelatinase activity were performed on modified Bennett’s agar medium (MBA) [28]. To determine cellulolytic efficiency, mineral salt agar supplemented with carboxymethyl cellulose (CMC) as the sole carbon source was employed [29]. The detection of secreted chitinase activity was performed in a chitinase test medium amended with colloidal chitin [30]. 

The most potent actinobacterial strains were molecularly identified. Genomic DNA extraction from each isolate was performed using the GeneJET plant genomic DNA purification kit (Thermo Scientific, Lithuania), following the manufacturer’s instructions. Subsequently, the DNA concentration was determined using Nanodrop One (Merc, Thermo Scientific). The prokaryotic universal primers, forward (27F: 5′-AGTT TGATCMTGGCTCAG-3′ and reverse 1492R: 5′-GGTTACCTTGTTACGACTT-3′), were employed to amplify the 16S rRNA gene sequence [31]. The PCR thermocycling parameters consisted of an initial denaturation at 95 °C for 3 min, followed by 32 cycles of denaturation at 94 °C for 30 s, annealing at 55 °C for 1 min, extension at 72 °C for 2 min, final extension at 72 °C for 10 min, and finally holding at 4 °C. The PCR reactions were executed using an Applied Biosystems 2720 thermal cycler. Subsequently, the amplified DNA products were electrophoretically separated on a 1.2% (*w*/*v*) agarose gel in 1×TBE buffer and visualized under a UV transilluminator using an EZ view. The molecular weights of the PCR products were estimated by comparison with the DNA marker weights VC 100 bp plus DNA ladder and VC 1 kbp plus DNA ladder. The purification and sequencing of the PCR products were performed by Macrogen (Macrogen, Inc., Korea). The obtained 16S rRNA gene sequences were aligned and compared with the NCBI GenBank database using the BLAST tool to extract homologous sequences for phylogenetic analysis. The phylogenetic tree was constructed using the neighbor-joining method with MEGA X software [32]. Bootstrap values (1000 repetitions) indicating the proportion of replicate trees in which connected taxa clustered together are presented next to the branches.

### 2.5. Effect of the Antagonistic Isolates on Sugar Beet Root Diseases in Greenhouse

The most potent actinobacteria that showed high inhibition and antagonistic effect against the phytopathogen *F. oxysporum* (F186) were evaluated for their effectiveness in controlling root rot disease in the greenhouse trial during the growing season 2021/2022.

#### 2.5.1. Preparation of Pathogen Inoculum

A pathogen inoculum was prepared by growing *F. oxysporum* on autoclaved sorghum grains in 500 mL glass bottles for 15 days at room temperature. The resulting growth of the pathogen was used to inoculate the sterile potted soil before planting. 

#### 2.5.2. Preparation of Microbial Suspensions

Actinobacterial suspensions were prepared and adjusted to 10^6^ CFU mL^−1^ as described by Errakhi et al. [33]. The seeds of three varieties, namely Toro c.v., Kwamera c.v., and Cleopatra c.v., were employed in this study. The seeds were rinsed with tap water and then surface sterilized with 5% sodium hypochlorite. Afterward, they were washed three times with sterile distilled water. Subsequently, seeds were coated separately by being soaked in the spore suspension of actinobacteria amended with Tween 80 solution (0.05%) for 4 h. Then, they were left to dry under a laminar flow hood. 

Thereafter, five seeds were cultivated immediately in each pot (no. 40) with three replicates. Seeds soaked in sterile distilled water acted as a control. On the other hand, seeds covered with the commercial fungicide Metalaxyl M + fludioxonil (1 cm/kg) served as a positive control. Metalaxyl M + fludioxonil was employed in this study due to its effectiveness against *Fusarium* species, causing the suppression of hyphal growth and adverse impacts on hyphal morphology, including cell wall breakdown, withering hyphae, and excessive septation. A randomized complete block was used in the experimental design. 

The applied treatments were categorized as follows: negative control: untreated and uninfected plants; infected control: plants infected with *F. oxysporum* F186 but left untreated; SB3-15+F: plants treated with *Streptomyces* strain SB3-15 and infected with *F. oxysporum* F186; SB2-23+F: plants treated with *Streptomyces* strain SB2-23 and infected with *F. oxysporum* F186; SB3-15: plants treated with SB3-15 but uninfected; SB2-23: plants treated with SB2-23 but uninfected; and fungicide: plants treated with the commercial fungicide Metalaxyl M + fludixonil (1 cm/kg) and infected with *F. oxysporum* F186.

### 2.6. Disease Assessments

The evaluation of pre- and post-emergence damping-off was conducted at two specific intervals, namely 15 days and 45 days after planting, following the methodology outlined by El-Argawy et al. [34].
$$Pre-emergence\;damping\;off\%=\frac{n}{B}\times 100$$
where *n* = the number of non-emerged seeds and *B* = the total number of seeds sown.
$$Post-emergence\;damping\;off\%=\frac{n}{E}\times 100$$
where *n* = the number of dead plants and *E* = the total number of emerging plants.

For each treatment and replicate, disease severity (DS) and disease incidence (DI) were evaluated on all sugar beet plants. The degree of root rot was determined according to [6] with the following rating scale: 0 = healthy sugar beet plant; 1 = mildly stunted to severely stunted plant, leaves may be wilted; 2 = leaves chlorotic, necrosis at leaf margins; 3 = crown drying out and becoming brown to black, leaves withering; and 4 = plant death. The following equation was used to compute the disease severity%: $$Disease\;Severity\%=\frac{\sum nb}{TM}\times 100$$
where *n* is the number of pathogenic plants at the same severity level, *b* represents the severity level, *T* represents the total number of examined plants, and *M* represents the maximum severity level.
$$Efficacy\%=\frac{Control-Treatment}{Control}\times 100$$

### 2.7. Growth Attributes and Yield Evaluation

Plants from each treatment were carefully uprooted and cleaned under running tap water 180 days after seeding. Thereafter, vegetative growth characteristics such as total fresh weight (gm), root weight (gm), root length (cm), and root diameter (cm) were assessed according to [35]. However, the sugar beet quality parameters (sucrose%, extractable sugar %, potassium %, α-amino nitrogen %, sodium %, and sugar loss to molasses %) were evaluated at Al-Fayoum sugar company laboratories.

### 2.8. Statistical Analysis

The data underwent statistical analysis using SPSS software for Windows version 25.0. Initially, all comparisons were assessed through a one-way analysis of variance (ANOVA) test. To identify significant differences among treatment means, Duncan’s multiple range test was employed at a significance level of *p* ≤ 0.05. Furthermore, the least significant difference (LSD) was utilized to evaluate variations between treatment means with a 5% probability level.

## 3. Results

### 3.1. Isolation of Actinobacteria

Thirty-seven morphologically different actinobacterial isolates were recovered from the rhizosphere of healthy sugar beet plants. Eight isolates were recovered from Fayoum, twelve from Beni-Suef, seven from Kafrelshikh, and ten from Giza governorates. All of the 37 actinobacterial isolates were screened for their potential to antagonize *F. oxysporum* (F186) in vitro. The results indicated that 15 out of 37 isolates showed an antagonistic effect against the target pathogen (Figure 1). Among them, two actinobacterial isolates marked with SB3-15 and SB2-23 were of particular interest because of their potent antagonistic activity against the target pathogen. They recorded the highest inhibition of the phytopathogen *F. oxysporum* compared to other isolated actinobacterial isolates (Figure 1). After dual culture with strains SB3-15 and SB2-23 for 7 days at 28 °C, *F. oxysporum* F186 showed narrow and oval colonies compared to the negative control (Figure 2). The mean mycelial growth diameter of F186 reached 8.50 cm in comparison to 4.41 cm of strain SB3-15 with an inhibitory rate of 48.24% and 3.2 cm of strain SB2-23 with an inhibitory rate of 62.35% (Figure 1). Therefore, the two strains were selected for further study.

### 3.2. Taxonomic Identification of Strains SB3-15 and SB2-23

Data in Table 1 show the morphological and cultural characteristics of the specific actinobacterial strains. Strain SB3-15 exhibited substantial to moderate growth across all ISP media employed. Its aerial mycelium color ranged from grayish white to grey, while the substrate mycelium color ranged from grayish white to grayish brown. No soluble pigments were observed on the utilized media. This strain formed spore chains organized in verticillate patterns.

In contrast, strain SB2-23 displayed robust growth on all media utilized. Its aerial mycelium color varied from grayish white to dark grey, and the substrate mycelium color ranged from yellowish white to grayish white. Notably, it produced a yellow pigment on ISP1 and ISP2. Additionally, it developed aerial mycelium characterized by short to long chains of coiled spores.

The enzymes and biochemical characteristics are given in Table 2, which showed that the two tested actinobacterial strains can produce cellulase, casein, gelatinase, catalase, and chitinase enzymes. Only strain SB3-15 can produce amylases; SB2-23 does not exhibit this ability. These observed traits align with typical morphological, physiological, and biochemical features commonly associated with *Streptomycete*-like organisms.

The identification process was strengthened by conducting a phylogenetic analysis using PCR-amplified and sequenced 16S rRNA sequences from SB3-15, SB2-23, and *Streptomyces* spp. This analysis confirmed that the two biologically active actinobacterial strains indeed belong to the genus *Streptomyces*, as depicted in Figure 3. The phylogenetic tree, generated through the neighbor-joining method, exhibited the placement of the two isolates into distinct clades within the tree structure. Notably, the tree strongly suggests that isolates SB3-15 and SB2-23 potentially signify a novel species within the genus *Streptomyces*. However, further taxonomic investigations are warranted to substantiate this proposed classification.

### 3.3. Effect of the Antagonistic Isolates on Sugar Beet Root Diseases in Greenhouse Disease Assessment 

The efficacy of the application of the two *Streptomyces* strains SB3-15 and SB2-23 on pre- and post-emergence damping-off and root rot of sugar beet varieties (cv., Toro; cv., Kwamera; cv., Cleopatra) caused by *F. oxysporum* was carried out in the greenhouse during the growing season of 2021/2022. Data in Table 3 show a significant reduction in damping-off disease due to the application of both *Streptomyces* strains, comparable to the effect of the chemical fungicide. Specifically, treatment with SB2-23 displayed the highest efficacy, notably reducing the average pre-emergence damping-off percentage to 6.67% compared to the 44.44% observed in the untreated infected control. Additionally, *Streptomyces* SB2-23 exhibited considerable effectiveness in reducing post-emergence damping-off caused by F186, achieving a notable reduction to 8.89%, surpassing the efficacy of other treatments tested and the infected control, which stood at 40%. 

Regarding root rot severity, observations from Table 4 demonstrate the absence of disease symptoms in the uninfected control, while all infected plants displayed typical Fusarium root rot symptoms. Notably, the infected, untreated control exhibited the highest severity of the disease, particularly evident in the Kwamera cv. (46.67%) compared to both treated plants and the uninfected control.

Treating the sugar beet plants with *Streptomyces* strains SB3-15 or SB2-23, alongside the chemical fungicide, resulted in a significant reduction in disease severity. There were no significant differences observed in disease severity among *Streptomyces* SB2-23, SB3-15, or the fungicide.

Interestingly, *Streptomyces* SB2-23 showcased the most prominent reduction in disease severity percentage (2.64%), displaying high protection with no disease symptoms (Figure 4). Recorded high efficacy rates of 94.28%, 91.06%, and 94.77% for Toro cv., Kwamera cv., and Cleopatra cv., respectively, in comparison to the fungicide treatment (92.36%, 86.44%, and 89.57%, respectively).

### 3.4. Sugar Beet Growth Attributes and Yield Evaluation

The findings depicted in Figure 5, Figure 6, Figure 7 and Figure 8 demonstrated that applying both *Streptomyces* strains, SB3-15 and SB2-23, whether in the presence or absence of the pathogen, led to a substantial increase in various growth parameters and yield metrics. Notably, there were significant enhancements observed in the fresh weight of roots, total weight of sugar beet plants, root length, and root diameter across all three varieties, compared to the infected and untreated control.

Remarkably, when focusing on the sugar beet Cleopatra variety, the application of SB2-23 showed statistically superior results across all studied growth characteristics. This signifies that SB2-23 had a notably more pronounced positive effect on growth parameters and yield, especially in the Cleopatra variety, outperforming both SB3-15 and the control groups in all aspects studied.

### 3.5. Sugar Beet Quality Traits

#### 3.5.1. Sucrose Percentage

The sucrose percentage analysis presented in Table 5 revealed the impact of various actinobacterial strains on the sucrose content within specific sugar beet varieties. The data indicated a significant rise in sucrose percentage due to the presence of actinobacterial strains SB3-15 and SB2-23. Conversely, the infected control treatment showcased the lowest sucrose values, followed by the negative control. Findings outlined in Table 5 highlighted substantial variations in sucrose percentage among the examined sugar beet varieties. Specifically, the Toro variety displayed the highest sucrose content, whereas the Cleopatra variety exhibited the lowest. The interaction between these factors notably influenced sucrose percentage. Remarkably, the sugar beet Toro variety treated with the SB2-23 strain achieved the highest sucrose percentage, reaching 17.7%.

#### 3.5.2. Extractable Sugar Percentage

Data outlined in Table 5 show that *Streptomyces* strains exhibited a statistical impact on the extractable sugar percentage, mirroring the influence observed on sucrose percentage. Sugar beet treated with actinobacterial strains, regardless of Fusarium infection, exhibited significantly higher extractable sugar percentages compared to both the negative and infected control treatments. Specifically, sugar beet treated with SB2-23 showed the highest extractable sugar content, followed closely by SB3-15, with no significant difference between them, but both significantly surpassing other treatments.

Additionally, the findings in Table 5 revealed significant differences among sugar beet varieties in terms of extractable sugar. Interestingly, the extractable sugar values seemed to align with high sucrose percentage varieties, with the Toro sugar beet variety registering the highest extractable sugar content.

The interaction between both factors (treatment with actinobacterial strains and sugar beet varieties) significantly influenced extractable sugar percentage. The behavior of the tested varieties regarding this trait followed a similar pattern to that observed for sucrose percentage under various treatments. Specifically, there was a significant contrast in extractable sugar percentage when the Toro and Cleopatra sugar beet varieties were treated with SB2-23 and SB3-15. However, there was no significant difference between them in their effect on extractable sugar.

#### 3.5.3. Impurities and Sugar Losses in Molasses

The data provided in Table 6 indicated that juice impurities, specifically potassium (K), sodium (Na), and alpha-amino nitrogen, showed insignificant differences across all treatments. However, there was a significant disparity in sugar loss to molasses among the treatments, with the infected control displaying the highest values for this trait.

Results in Table 6 manifested that impurities and sugar loss to molasses differed significantly among sugar beet varieties. It was clear that the Toro variety exhibited the lowest Na and K (4.1 and 2.3 mg/100 g beet), while the lowest sugar loss to molasses (1.9 mg/100 g beet) was recorded by Kwamera. On the other hand, the highest values of alpha-amino nitrogen were in the Toro and Cleopatra varieties, respectively. 

The interaction between both factors (treatment with actinobacterial strains and sugar beet varieties) significantly influenced juice impurities (K, Na, and α-amino nitrogen) and sugar loss to molasses.

## 4. Discussion 

Fusarium root rot affecting sugar beet is recognized as one of the most detrimental soil-borne fungal diseases globally, significantly impeding the sustainable advancement of the sugar industry [7,36]. The extensive use of chemical fungicides has resulted in severe concerns related to human health and environmental pollution. Consequently, biological control strategies are being explored as viable and sustainable alternatives [37]. Actinobacteria exhibit a remarkable capacity to counteract the harmful impacts associated with synthetic fungicides while also demonstrating positive effects on plant growth, underscoring their crucial role in maintaining ecosystem resilience [38]. Over recent decades, *Streptomyces* has emerged to act as a weapon, serving as a defense mechanism against phytopathogen infections and enhancing crop yield [21,39,40]. In this study, of the 37 actinobacterial isolates obtained from sugar beet rhizosphere soil, 15 exhibited significant antifungal activity against the tested phytopathogen *F. oxysporum* F186. This finding highlights the potential utilization of these isolates for controlling phytopathogenic diseases. The antagonistic activity of actinobacteria against phytopathogens can be achieved through diverse mechanisms such as antibiosis, nutrient competition, the release of degradative enzymes, induced resistance, and the production of nitrous oxide [13,41]. In our study, we assessed the antagonistic effects of isolated actinobacteria through antibiosis, wherein secondary metabolites from actinobacterial strains diffuse in the agar medium, inhibiting the growth of the *F. oxysporum* pathogen. Notably, actinobacterial strains SB3-15 and SB2-23 exhibited the most potent inhibition of growth against *F. oxysporum* F186.

The morphological and physiological characterization of these selected strains demonstrated their capability to generate various hydrolytic enzymes like cellulases, amylases, proteases, gelatinases, catalases, and chitinases. The antifungal mechanism of these strains might be associated with their ability to produce such hydrolytic enzymes. Even though strain SB2-23 did not exhibit amylase activity, it displayed the superior inhibition of *F. oxysporum* growth. This outcome underscores the significant role of chitinases in the biocontrol behavior of *Streptomyces* against root rot pathogens in sugar beet, as previously discussed by Karimi et al., 2012 [42]. Moreover, these outcomes are consistent with findings from other researchers [43,44], who documented the correlation between the biocontrol efficacy of various *Streptomyces* species and the production of hydrolytic enzymes like chitinases, proteases, and cellulases. These enzymes are known to disrupt the cell walls of fungi, leading to the leakage of cellular contents and subsequently inhibiting the growth of phytopathogenic fungi [45]. Multiple previous studies have also highlighted the potential of actinobacteria to influence the growth of *Fusarium* spp. by generating hydrolytic enzymes [46,47,48,49,50,51]. Moreover, *Streptomyces* possess the capacity to inhabit plant root surfaces, survive in diverse soil types, and generate spores, enabling their prolonged existence even in varying extreme conditions [52]. *Streptomyces* strains exhibit a wide array of antibiotics and volatile organic compounds that function against pathogens, disrupting fungal cell-to-cell communication. Additionally, they produce an array of enzymes that degrade the cell walls of fungi [19,53,54].

However, the capability of a strain to suppress disease in vitro may not necessarily translate to its effectiveness as a biocontrol agent in vivo, as the strain may not demonstrate its potential in natural circumstances [55]. Therefore, in vivo pot experiments were conducted under natural conditions to evaluate the effectiveness of employing *Streptomyces* strains SB3-15 and SB2-23 as biocontrol agents against root rot disease triggered by *F. oxysporum*. Damping-off disease, induced by *F. oxysporum* F186, hinders seed germination and early seedling emergence, leading to the decay of germinating seeds and young seedlings. Our findings demonstrated that *Streptomyces* strains SB3-15 and SB2-23 exhibited substantial efficacy in reducing the number of damping-off occurrences and enhancing plant survival rates. Additionally, they lessened disease severity compared to the infected control by directly impacting the pathogen and positively influencing seed germination rates [56]. Moreover, isolate SB2-23 exhibited the highest protective activity with high efficacy ranging from 91.06 to 94.77% compared to chemical fungicide (86.44 to 92.36%). Several *Streptomyces* spp. and their bioactive substances have been demonstrated to be effective biocontrol agents against a variety of plant diseases [46,57,58,59]. Recent research has highlighted the antimicrobial properties of phenolic compounds, such as phenol 2,4-bis(1,1-dimethylethyl) and phenol 2,2- methylenebis[6-(1,1-dimethylethyl)-4-methyl], extracted from *Streptomyces* strain H3-2 due to their ability to eliminate free radicals. Additionally, compounds like alkane (4,6- dimethyldodecane) and olefin (benzene, 1,2,3- trimethoxy-5-(2- propenyl)-) identified in the GC-MS fractions have demonstrated strong antimicrobial activity, showcasing the potent biocontrol efficiency of *Streptomyces* sp. H3-2 against banana Fusarium wilt disease [15].

Furthermore, the present study revealed that the application of the two *Streptomyces* strains, SB3-15 and SB2-23, significantly enhanced multiple growth attributes in sugar beet. This enhancement was observed in increased total weight, root fresh weight, root diameter, and root length when compared to negative and infected controls. Additionally, these applications exhibited heightened resistance against Fusarium root rot. These findings align with prior research [60], which supports the hypothesis that the plant-associated *Streptomycetes* give a variety of plant benefits and enhanced root weight, shoot, and root length of sugar beet compared to the non-inoculated treatments. Additionally, the enhancement observed in sugar beet growth attributes corroborates with findings reported in several earlier studies [61,62]. *Streptomyces* are acknowledged for their production of phytohormones such as ethylene, gibberellic acid, indole-3-acetic acid (IAA), phenylacetic acid, cytokines, and ACC deaminases [18,40]. Several phytohormones perform critical roles in controlling the growth of plant parts at various stages of development [63]. PGPRs may synthesize phytohormones comparable to those synthesized by plants, which aid in plant development [64]. Furthermore, they participate in plant defense responses against numerous diseases [65,66]. IAA synthesized by *Streptomyces* strains increased root length and surface area, which enhances nutrient absorption from the soil [13]. However, in all growth variables evaluated, the sugar beet Cleopatra variety treated with SB2-23 outperformed the two varieties statistically. The improvement in sugar beet root growth features caused by the application of SB2-23 and/or SB3-15 might be related to these strains’ crucial function in controlling root rot and boosting plant development [67,68].

In addition, it can be noted that the distinct effect of SB3-15 and SB2-23 strains on the sucrose percentage of sugar beet varieties is somewhat growth promoting, as it accelerates seed germination, closer canopy, plant growth to reach the storage stage earlier, which is considered more suitable for sugar translocation and accumulation [69,70]. Sugar beet varieties, on the other hand, fluctuate greatly in sucrose percentage content. Toro sugar beet variety achieved a higher sucrose% value. Meanwhile, the Cleopatra variety had the lowest value of this feature. This finding satisfied researchers that this trait is strongly correlated with genetic make-up [71,72]. Furthermore, treated sugar beet with SB2-23 in the presence or absence of soil infestation with *F. oxysporum* F186 attained the highest extractable sugar followed by SB2-15 without significance between them; however, both of them statistically surpassed the other treatments. This result is in agreement with [71,73]. Moreover, juice impurities, i.e., K, Na, and alpha amino nitrogen and sugar loss to molasses differed significantly among sugar beet varieties. Toro variety exhibited the lowest Na and K, while the lowest sugar loss to molasses was recorded by Kwamera. On the other hand, the highest values of alpha-amino nitrogen were of Toro and Cleopatra varieties, respectively. Varieties may be differed in their nutrient absorption ability reflecting the genetic makeup among varieties [74]. In this context, previous studies, namely [71,75], indicated noticeable variations among beet varieties in terms of their quality attributes. The application of *Streptomyces* SB2-15 and SB2-23 resulted in a significant reduction of alpha-amino N across all varieties, especially in Kwamera. This reduction is crucial as elevated levels of alpha-amino N in storage roots can hinder sucrose extractability [76,77]. These findings indicate a potential future use of *Streptomyces* strains as effective agents for biological control against pathogenic fungi.

## 5. Conclusions

This study reveals the efficacy of *Streptomyces* sp. SB3-15 and SB2-23 isolated from sugar beet rhizosphere in controlling Fusarium root rot. The strains displayed notable effectiveness in controlling root rot and diminishing its severity in three distinct sugar beet varieties (Toro, Kwamera, and Cleopatra). Moreover, strain SB2-23 exhibited superiority, enhancing growth parameters, yield, sugar content, and purity across all three cultivars. These findings suggest that utilizing *Streptomyces* strain SB2-23 offers both biocontrol efficacy against Fusarium root rot and growth-enhancing effects on sugar beet plants, presenting a safe and eco-friendly solution.

## Figures and Tables

**Figure 1 plants-13-00311-f001:**
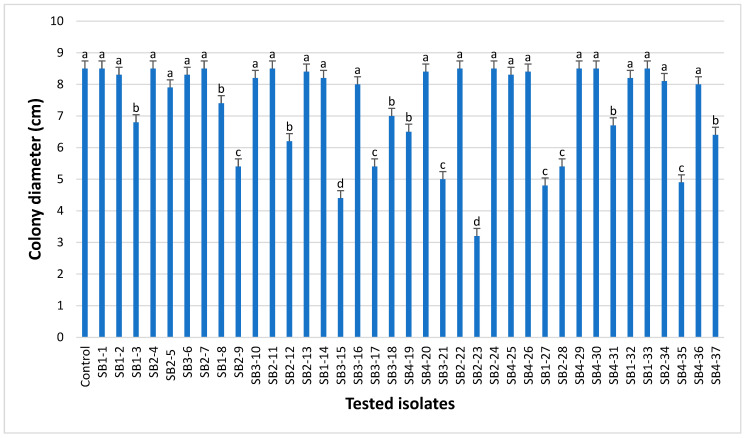
The in vitro antagonistic activity of *Streptomyces* strains against *F. oxysporum* (F186), measured as growth diameter in control and each treatment. Different lowercase letters represent a significant difference according to Duncan’s multiple range test (*p* < 0.05).

**Figure 2 plants-13-00311-f002:**
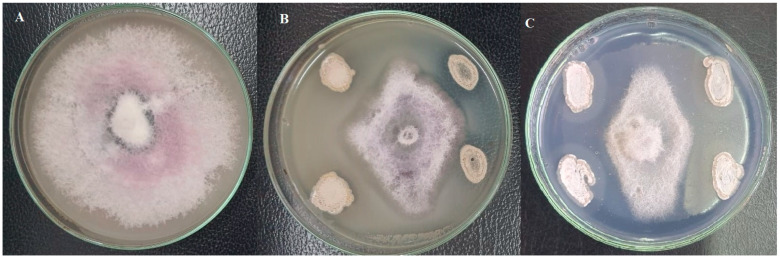
Antagonistic activity of strains SB3-15 and SB2-23 against *F. oxysporum*. (**A**) *F. oxysporum* (F186) grew on the PDA plate alone, 7 days after inoculation. (**B**) Strain SB3-15, inhibiting the mycelial growth of *F. oxysporum* F186. (**C**) Strain SB2-23, inhibiting the mycelial growth of *F. oxysporum* F186.

**Figure 3 plants-13-00311-f003:**
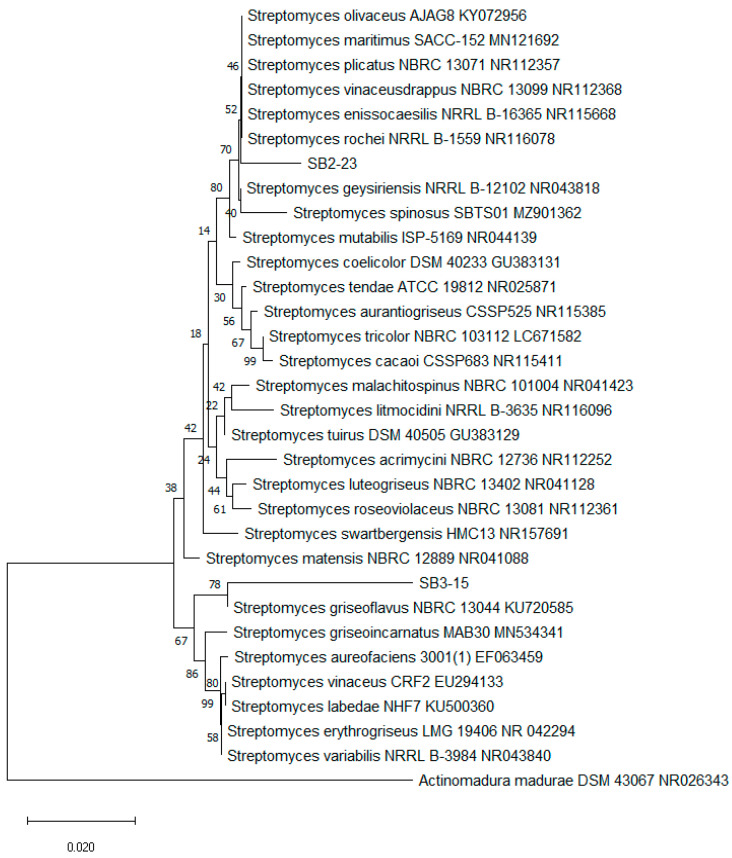
A neighbor-joining phylogenetic tree constructed using 16S rRNA gene sequences demonstrating the connections between the two selected isolates (SB3-15 and SB2-23) and other closely associated *Streptomyces* members. The node numbers are percent bootstrap levels based on 1000 replications of reconfigured datasets. Actinomadura madurae DSM 43067 NR026343 is used as an outgroup.

**Figure 4 plants-13-00311-f004:**
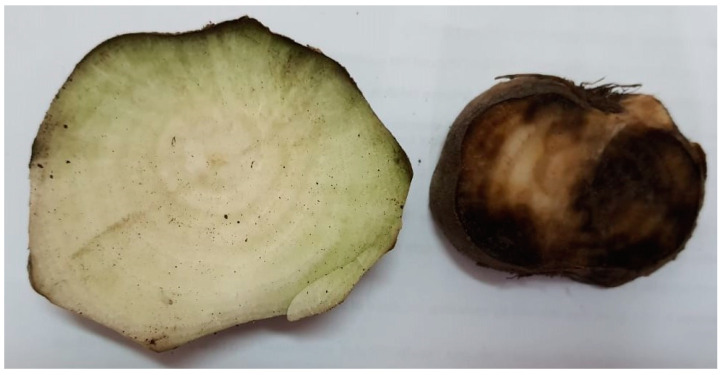
Vascular tissues of sugar beet roots show the effect of strain SB2-23 fermentation broth on the growth of sugar beet and resistance to *F. oxysporum* (**left**) and the infected control exhibited grayish-brown to black discoloration due to infection with *F. oxysporum* (**right**).

**Figure 5 plants-13-00311-f005:**
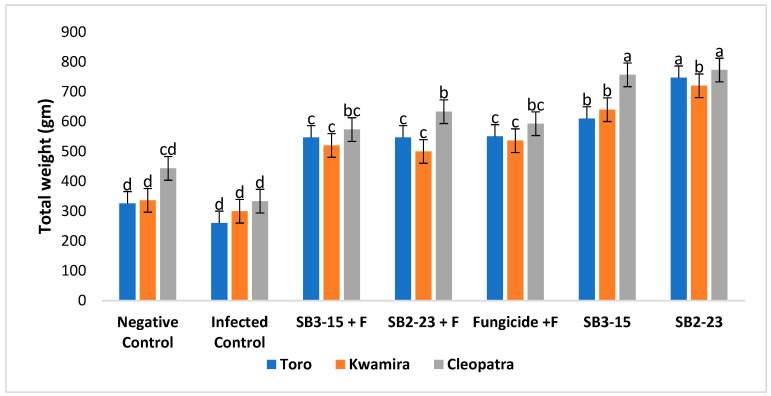
Assessing the impact of *Streptomyces* strains on the total weight of sugar beet varieties. Negative control: uninfected and untreated control; infected control: infected with *F. oxysporum* F186 and untreated; SB3-15+F: treated with *Streptomyces* strain SB3-15 and infected with *F. oxysporum* F186; SB2-23+F: treated with *Streptomyces* strain SB2-23 and infected with *F. oxysporum* F186; SB3-15: uninfected, treated with strain SB3-15; SB2-23: uninfected, treated with strain SB2-23; fungicide: infected with F186 and treated with the commercial fungicide Metalaxyl M + fludixonil (1 cm/kg). The data are provided as the mean of at least three replicates ± standard error. Different little letters (a–d) above the bars denote significant differences according to Duncan’s multiple range test at *p* ≤ 0.05.

**Figure 6 plants-13-00311-f006:**
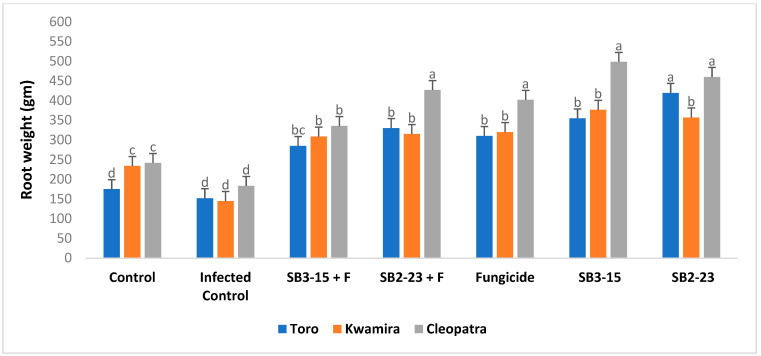
Assessing the impact of *Streptomyces* strains on the fresh root weight of sugar beet varieties. Negative control: uninfected and untreated control; infected control: infected with *F. oxysporum* F186 and untreated; SB3-15+F: treated with *Streptomyces* strain SB3-15 and infected with *F. oxysporum* F186; SB2-23+F: treated with *Streptomyces* strain SB2-23 and infected with *F. oxysporum* F186; SB3-15: uninfected, treated with strain SB3-15; SB2-23: uninfected, treated with strain SB2-23; fungicide: infected with F186 and treated with the commercial fungicide Metalaxyl M + fludixonil (1 cm/kg). The data are provided as the means of at least three replicates ± standard error. Different little letters (a–d) above the bars denote significant differences according to Duncan’s multiple range test at *p* ≤ 0.05.

**Figure 7 plants-13-00311-f007:**
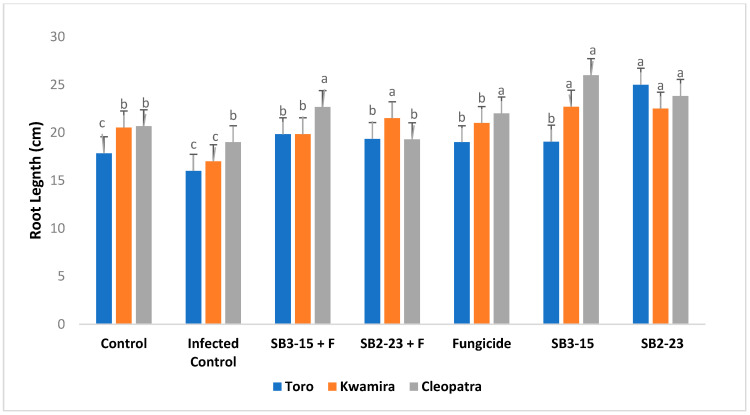
Assessing the impact of *Streptomyces* strains on the root length of sugar beet varieties. Negative control: uninfected and untreated control; infected control: infected with *F. oxysporum* F186 and untreated; SB3-15+F: treated with *Streptomyces* strain SB3-15 and infected with *F. oxysporum* F186; SB2-23+F: treated with *Streptomyces* strain SB2-23 and infected with *F. oxysporum* F186; SB3-15: uninfected, treated with strain SB3-15; SB2-23: uninfected, treated with strain SB2-23; fungicide: infected with F186 and treated with the commercial fungicide Metalaxyl M + fludixonil (1 cm/kg). The data are provided as the means of at least three replicates ± standard error. Different little letters (a–c) above the bars denote significant differences according to Duncan’s multiple range test at *p* ≤ 0.05.

**Figure 8 plants-13-00311-f008:**
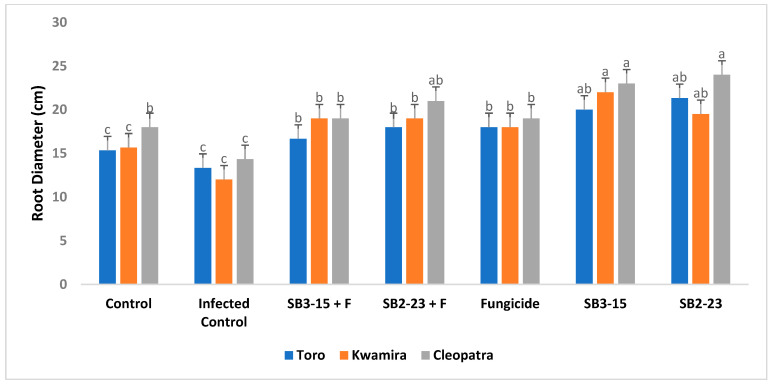
Assessing the impact of *Streptomyces* strains on the root diameter of sugar beet varieties. Negative control: uninfected and untreated control; infected control: infected with *F. oxysporum* F186and untreated; SB3-15+F: treated with *Streptomyces* strain SB3-15 and infected with *F. oxysporum* F186; SB2-23+F: treated with *Streptomyces* strain SB2-23 and infected with *F. oxysporum* F186; SB3-15: uninfected, treated with strain SB3-15; SB2-23: uninfected, treated with strain SB2-23; fungicide: infected with F186 and treated with the commercial fungicide Metalaxyl M + fludixonil (1 cm/kg). The data are provided as the means of at least three replicates ± standard error. Different little letters (a–c) above the bars denote significant differences according to Duncan’s multiple range test at *p* ≤ 0.05.

**Table 1 plants-13-00311-t001:** Morphological and cultural traits of actinobacterial strains SB3-15 and SB2-23.

Characteristics	SB3-15	SB2-23
**Tryptone yeast extract agar (** **ISP1)**		
Proliferation	Abundant	Abundant
Above mycelium color	Grayish brown	Grayish white
Basal mycelium color	Grayish white	Yellowish white
Soluble Pigmentation	None	Lemon yellow
**Malt extract yeast extract dextrose agar (ISP2)**		
Proliferation	Abundant	Abundant
Above mycelium color	Grey	Grayish white
Basal mycelium color	Dark brown	Yellowish white
Soluble Pigmentation	None	Light yellow
**Inorganic salts starch agar (ISP4)**		
Proliferation	Good	Abundant
Above mycelium color	Grayish white	Dark grey
Basal mycelium color	Grayish brown	Grayish white
Soluble Pigmentation	None	None
**Peptone yeast extract iron agar (ISP6)**		
Proliferation	Moderate	Good
Above mycelium color	Grayish yellow	Light grey
Basal mycelium color	Yellowish brown	Grayish white
Soluble Pigmentation	None	None
**Potato dextrose agar (PDA)**		
Proliferation	Abundant	Abundant
Above mycelium color	Grayish white	Dark grey
Basal mycelium color	Grayish brown	Pale yellow
Soluble Pigmentation	None	None

**Table 2 plants-13-00311-t002:** Biochemical characteristics of actinobacteria strains SB3-15 and SB2-23.

Actinobacteria Isolate	CMC	Casein	Amylase	Gelatinase	Chitinase	Catalase
SB3-15	+	+	+	+	+	+
SB2-23	+	+	−	+	+	+

**Table 3 plants-13-00311-t003:** Evaluation of *Streptomyces* strains SB3-15 and SB2-23 potency on controlling pre- and post-emergence damping-off of sugar beet varieties caused by *F. oxysporum* F186 under greenhouse conditions.

Treatment	Pre-Emergence %				Post-Emergence %			
	V1	V2	V3	Mean	V1	V2	V3	Mean
SB3-15+F	16.67	6.67	20.00	14.45 ^b^	26.67	20.00	16.67	21.11 ^b^
SB2-23+F	13.33	6.67	0.00	6.67 ^c^	13.33	6.67	6.67	8.89 ^c^
SB3-15	0.00	0.00	0.00	0.00 ^d^	0.00	0.00	0.00	0.00 ^d^
SB2-23	0.00	0.00	0.00	0.00 ^d^	0.00	0.00	0.00	0.00 ^d^
Fungicide	13.33	20.00	6.67	13.33 ^b^	33.33	20.00	20.00	24.44 ^b^
Infected control	43.33	46.67	43.33	44.44 ^a^	40.00	33.33	46.67	40.00 ^a^
Negative control	0.00	6.67	0.00	2.22 ^c^	0.00	0.00	0.00	0.00 ^d^
Mean	12.38	12.38	10.00		16.19	11.43	12.86	

V1: cv., Toro; V2: cv., Kwamera; V3: cv., Cleopatra. SB3-15+F: treated with *Streptomyces* strain SB3-15 and infected with *F. oxysporum* F186; SB2-23+F: treated with *Streptomyces* strain SB2-23 and infected with *F. oxysporum* F186; SB3-15: uninfected, treated with strain SB3-15; SB2-23: uninfected, treated with strain SB2-23; fungicide: infected and treated with the commercial fungicide Metalaxyl M + fludixonil (1 cm/kg); infected control: infected with *F. oxysporum* F186 and untreated; negative control: uninfected and untreated control. Means within a column followed by a different letter(s) are significantly different according to Duncan’s multiple range test at *p* ≤ 0.05. Values are the means of three replicates for each treatment as well as the control.

**Table 4 plants-13-00311-t004:** Evaluation of *Streptomyces* strains SB3-15 and SB2-23 potency on controlling root rot disease of sugar beet varieties caused by *F. oxysporum* F186 under greenhouse conditions.

Treatment	Disease Severity %				Efficacy%			
	V1	V2	V3	Mean	V1	V2	V3	Mean
SB3-15+F	6.94	9.72	5.55	7.40 ^b^	80.93	79.17	82.62	80.91
SB2-23+F	2.08	4.17	1.67	2.64 ^b^	94.28	91.06	94.77	93.37
SB3-15	0.00	0.00	0.00	0.00 ^c^	-	-	-	-
SB2-23	0.00	0.00	0.00	0.00 ^c^	-	-	-	-
Fungicide	2.78	6.33	3.33	4.15 ^b^	92.36	86.44	89.57	89.46
Infected control	36.39	46.67	31.94	38.33 ^a^	0	0	0	0
Negative control	0.00	0.00	0.00	0.00 ^c^	-	-	-	-
Mean	6.88	9.56	6.07		89.19	85.56	88.99	

V1: cv., Toro; V2: cv., Kwamera; V3: cv., Cleopatra. (-): no efficacy. SB3-15+F: treated with *Streptomyces* strain SB3-15 and infected with *F. oxysporum* F186; SB2-23+F: treated with *Streptomyces* strain SB2-23 and infected with *F. oxysporum* F186; SB3-15: uninfected, treated with strain SB3-15; SB2-23: uninfected, treated with strain SB2-23; fungicide: infected and treated with the commercial fungicide Metalaxyl M + fludixonil (1 cm/kg); infected control: infected with *F. oxysporum* F186 and untreated; negative control: uninfected and untreated control. Means within a column followed by a different letter(s) are significantly different according to Duncan’s multiple range test at *p* ≤ 0.05. Values are the means of three replicates for each treatment as well as the control.

**Table 5 plants-13-00311-t005:** The interaction between *Streptomyces* treatment and sugar beet varieties on percentage of sucrose and extractable sugar.

Treatments	Sucrose %				Extractable Sugar %			
	V1	V2	V3	Mean	V1	V2	V3	Mean
Negative control	14.4	13.8	14.4	14.2	11.8	11.2	11.5	11.5
Infected control	12.4	11.8	12.2	12.1	10.2	10.8	10.1	10.4
SB3-15+F	15.8	14.7	15.1	15.2	12.4	12.2	12.3	12.3
SB2-23+F	14.8	15.9	14.8	15.2	12.3	12.1	11.9	12.1
SB3-15	16.5	16.8	16.5	16.6	13.6	12.9	12.7	13.1
SB2-23	17.7	17.3	17.1	17.4	13.5	12.7	13.3	13.2
Fungicide + F	15.4	15.6	14	15	12.2	12.2	12.1	12.2
Mean	15.3	15.1	14.9		12.3	12	12	
LSD at 0.05								
Treatments (T)				1.02				0.019
Varieties (V)				0.31				0.037
T × V				0.055				0.269

V1: cv., Toro; V2: cv., Kwamera; V3: cv., Cleopatra. Negative control: uninfected and untreated control; infected control: infected with *F. oxysporum* F186 and untreated; SB3-15+F: treated with *Streptomyces* strain SB3-15 and infected with *F. oxysporum* F186; SB2-23+F: treated with *Streptomyces* strain SB2-23 and infected with *F. oxysporum* F186; SB3-15: uninfected, treated with strain SB3-15; SB2-23: uninfected, treated with strain SB2-23; fungicide + F: infected and treated with the commercial fungicide Metalaxyl M + fludixonil (1 cm/kg).

**Table 6 plants-13-00311-t006:** The interaction between *Streptomyces* treatment and sugar beet varieties on percentage of K, Na, α- amino N, and molasses.

Treatments	SLM%				Na				K				α- Amino N			
	V1	V2	V3	Mean	V1	V2	V3	Mean	V1	V2	V3	Mean	V1	V2	V3	Mean
Negative control	1.7	2.1	2	1.9	4	4.5	4.6	4.4	2.2	2	2	2.1	2.3	2.6	2.2	2.3
Infected control	2	2.1	2.1	2.1	4.4	4.8	4.5	4.6	2.1	2.3	1.8	2.1	2.5	2.3	2.9	2.6
SB3-15+F	2	1.9	2.1	2	3.7	4.4	4.6	4.2	2.9	2.3	2.2	2.5	2.1	1.8	2.6	2.2
SB2-23+F	2	1.9	1.9	1.9	4.2	3.7	4.2	4.1	2	2.6	2.2	2.3	2.4	2	2.1	2.2
SB3-15	1.9	2	1.9	1.9	4.2	4.2	4	4.1	2.2	3	2.4	2.5	2.1	1.8	2.2	2
SB2-23	2.1	2	2	2	4.6	4.2	4.3	4.3	2	2.6	2.6	2.4	2.8	2.1	2.1	2.3
Fungicide + F	2.2	1.7	2	2	4	3.5	3.9	3.8	2.3	2	2.6	2.3	2.5	2.4	2.2	2.4
Mean	2	1.9	2		4.1	4.2	4.3		2.3	2.4	2.3		2	1.8	2	
LSD at 0.05																
Treatments (T)				0.021				NS				NS				NS
Varieties (V)				0.029				0.16				0.09				0.21
T × V				0.078				NS				NS				NS

V1: cv., Toro; V2: cv., Kwamera; V3: cv., Cleopatra. Negative control: uninfected and untreated control; infected control: infected with *F. oxysporum* F186 and untreated; SB3-15+F: treated with *Streptomyces* strain SB3-15 and infected with *F. oxysporum* F186; SB2-23+F: treated with *Streptomyces* strain SB2-23 and infected with *F. oxysporum* F186; SB3-15: uninfected, treated with strain SB3-15; SB2-23: uninfected, treated with strain SB2-23; fungicide + F: infected and treated with the commercial fungicide Metalaxyl M + fludixonil (1 cm/kg).

## Data Availability

Data is contained within the article.

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
