# Peer review of "Combatting Sugar Beet Root Rot: Streptomyces Strains’ Efficacy against Fusarium oxysporum"

_plants, 2024, doi:10.3390/plants13020311_

Round 1

Reviewer 1 Report

Comments and Suggestions for Authors

1. 37 isolates were got from  different governorates of Egypt. How many were got from where,more information is needed. 

2. Seeds coated with commercial fungicide Metalaxyl M + fludixonil (1 cm/ kg) act as positive control. Fungicide Metalaxyl M has no inhibitory activity against Fusarium.

3. 2.5 more information is needed for treatments, and how treated?

4. the discussion should be focus for the results and conclusion should be shortened.

5. other comments are attached.

Author Response

Dear Reviewer#

Thank you very much for spending the time to review our manuscript. Please find the detailed responses below and the corresponding revisions/corrections highlighted/in track changes in the re-submitted files.

Comments 1: 37 isolates were got from  different governorates of Egypt. How many were got from where, more information is needed. 

Response 1: Thank you for pointing this out. We agree with this comment. Therefore, we have put the data of isolation and the number of isolates from each governorate in the results section.

Comments 2. Seeds coated with commercial fungicide Metalaxyl M + fludixonil (1 cm/ kg) act as positive control. Fungicide Metalaxyl M has no inhibitory activity against Fusarium.

Response 2: Thank you but we don’t agree with this comment because Metalaxyl M + fludixonil is effective and has inhibitory activity against Fusarium. Herein are some articles that emphasize this issue.

Miguel Tde Á, Bordini JG, Saito GH, Andrade CG, Ono MA, Hirooka EY, Vizoni É, Ono EY. Effect of fungicide on Fusarium verticillioides mycelial morphology and fumonisin B₁ production. Braz J Microbiol. 2015 Mar 31;46(1):293-9. doi: 10.1590/S1517-838246120120383. PMID: 26221120; PMCID: PMC4512075.

José Antonio Rodríguez-Liébana, Alberto López-Galindo, Concepción Jiménez de Cisneros, Antonia Gálvez, Marisa Rozalén, Rita Sánchez-Espejo, Emilia Caballero, Aránzazu Peña, Adsorption/desorption of fungicides in natural clays from Southeastern Spain, Applied Clay Science, 132–133, 2016, 402-411. https://doi.org/10.1016/j.clay.2016.07.006.

Sameer, W. M. Compatibility of biological control agents with fungicides against root rot diseases of wheat. Al-Azhar Journal of Agricultural Research. 2: (44), 2019. 146-155. https://doi.org/10.21608/AJAR.2019.102808.

Comments 3. 2.5 more information is needed for treatments, and how treated?

Response 3: Need full done

Comments 4. the discussion should be focus for the results and conclusion should be shortened.

Response 4: Needful done, the discussion was rewritten for better presentation, and was significantly focused for the results. Conclusion was reduced and shortened.

Comments 5:  other comments are attached.

Response 5: All comments found in the attached file were fully done

All corrections and modifications are in the attached file

Reviewer 2 Report

Comments and Suggestions for Authors

In this paper the authors assessed some Streptomyces strains for effectiveness in the biocontrol of Fusarium oxysporum-Induced Root Rot in Sugar Beet. The article is well-organized and clearly presents the background and context of the study. The results are convincing. Overall, the study is well-presented and informative. A major concern of mine is that there are many tables in the text. It is not intuitive to show results with tables. Pls transfer them into figures. In addition, Include specific references to the literature throughout the discussion to support your statements. For instance, when discussing the potential of Streptomyces as a biocontrol agent, refer to relevant studies that have demonstrated similar findings. 

Specify the criteria used for selecting the two strains (SB3-15 and SB2-23) for further study. What specific characteristics or activities made them of particular interest?

When discussing the impact of Fusarium species on sugar beet, providing specific examples or statistics, if available, could enhance the understanding of the severity of the issue.

Provide a brief description or context for readers who might not be familiar with the term "damping-off disease." Explain what it involves and why it is significant in the context of sugar beet cultivation.

minor comments

L14-15, L28-29: improve the sentences. 

L231 include all data that 37 isolates inhibating Fusarium in Fig. 2

L233 taxomomic identification

Line# 100. Check for any grammatical errors or awkward phrasing.

Section 2.8. Statistical Analysis. Provide a brief explanation of the statistical analyses performed, such as ANOVA and Duncan's Multiple Range test, to help readers understand how the data were analyzed e.g.  How many replication were used.

Comments on the Quality of English Language

The language is generally ok, but pay attention to the minor comments I have made.

Author Response

Dear Reviewer#

Thank you very much for spending the time to review our manuscript. Please find the detailed responses below and the corresponding revisions/corrections highlighted/in track changes in the re-submitted files.

Comments 1: In this paper the authors assessed some Streptomyces strains for effectiveness in the biocontrol of Fusarium oxysporum-Induced Root Rot in Sugar Beet. The article is well-organized and clearly presents the background and context of the study. The results are convincing. Overall, the study is well-presented and informative. A major concern of mine is that there are many tables in the text. It is not intuitive to show results with tables. Pls transfer them into figures. In addition, Include specific references to the literature throughout the discussion to support your statements. For instance, when discussing the potential of Streptomyces as a biocontrol agent, refer to relevant studies that have demonstrated similar findings. 

Response 1: Thank you for your pointing. About tables, authors wish to keep tables as they clearly present the data and the interaction between sugar beet varieties and treatments.

Comments 2: Specify the criteria used for selecting the two strains (SB3-15 and SB2-23) for further study. What specific characteristics or activities made them of particular interest?

Response 2: Need fully done, section 3.1 in Results

Comments 3: When discussing the impact of Fusarium species on sugar beet, providing specific examples or statistics, if available, could enhance the understanding of the severity of the issue.

Response 3: Included in the introduction section L39-51 with references that indicated the impact of Fusarium species on sugar beet yield, sucrose percentage and juice purity

Comments 4: Provide a brief description or context for readers who might not be familiar with the term "damping-off disease." Explain what it involves and why it is significant in the context of sugar beet cultivation.

Response 4: Agree with your comment. We have, accordingly, done and included it in the Discussion section  “Damping-off disease, caused by F. oxysporum, prevents seeds from germinating or seedlings from emerging, resulting in the destruction and rotting of germinating seeds and early seedlings.”

Comments 5: L14-15, L28-29: improve the sentences. 

Response 5: Need fully done

Comments 6: L231 include all data that 37 isolates inhibating Fusarium in Fig. 2

Response 6: Need fully done, the 37 isolates included and changed to be Fig. 1  to fit with the writing sequence

Comments 7: L233 taxomomic identification

Response 7: Need fully done

Comments 8: Line# 100. Check for any grammatical errors or awkward phrasing.

Response 8: revised and modified

 Comments 9: Section 2.8. Statistical Analysis. Provide a brief explanation of the statistical analyses performed, such as ANOVA and Duncan's Multiple Range test, to help readers understand how the data were analyzed e.g.  How many replication were used.

Response 9: included in L173 “three replicates”

Comments 10: The language is generally ok, but pay attention to the minor comments I have made.

Response 10: All comments on language was taken into consideration and modified

All corrections and modifications are in the attached file

Reviewer 3 Report

Comments and Suggestions for Authors

The introduction provides a comprehensive overview of the importance of sugar beet in Egypt, the threats posed by Fusarium species to sugar beet cultivation, and the potential of Streptomyces isolates as biocontrol agents. It ss well-structured and informative, with a strong emphasis on the significance of the research.

Matherial and methods:

The methodology is strong in its overall structure and detail but could benefit from additional explanations in certain areas and improved clarity.

Line 88: Clarify Sampling Locations and Number of Samples
Line 102: The text mentions incubating plates at 30°C for 15 days. It would be beneficial to briefly explain why this temperature and duration were chosen, linking it to the optimal conditions for the growth of actinobacteria.
Line 105: The description of long-term storage in 15% glycerol at -80°C is clear. However, mentioning the rationale for choosing glycerol as the cryoprotectant and the typical duration for which cultures can be stored would be informative.
Line 108: Citation?
Line 110: The number identifying the strain?
Line 111 and further: PDA provider?
Line 166: name of the authors of the methodology cited
Line 174:
Clarify the specific purpose of using commercial fungicide Metalaxyl M + fludixonil (1 cm/kg) as the positive control. Explain why this particular fungicide was chosen.
Line 184: Briefly explain the significance of these specific time points in relation to sugar beet growth stages.
Results:
Table 2 Can those results be quantified to compare the actual quantity of produced enzymes?
Table 3 and 4: SD should be added.
Table 5-6: statistical analysis is needed
Discussion:

Provide more detailed information on specific findings, such as the morphological and physiological characterization of the selected strains and the role of specific hydrolytic enzymes. Where applicable, consider quantifying the results to provide readers with a more concrete understanding of the observed effects, particularly in relation to disease severity, growth attributes, and sugar beet quality. Emphasize the novelty or unique contributions of the current study, especially if there are aspects that set it apart from existing literature.

The conclusion is concise but could be more engaging. It should summarize the key findings and emphasize the broader implications of the study.
The literature could include more positions from the last 5 years.

The experiment conducted in this study is relatively straightforward and may not introduce a novel approach to the field. However, with some enhancements, it has the potential to be of interest to readers. By refining the experimental design, providing more detailed quantitative results, and offering a clearer presentation of the findings, the study could become a valuable contribution. Additionally, the article could benefit from highlighting the practical implications of the research and how the improved methodology may have applications or significance in relevant contexts. While the initial experiment may be considered basic, the suggested improvements can elevate its relevance and appeal to a broader audience.

Comments on the Quality of English Language

The English used in the article is generally clear and comprehensible, but there are some areas where improvements can be made for clarity and precision. Some sentences are lengthy and might benefit from being broken down into shorter, more concise statements. This can enhance readability and comprehension. The organization of the article must be checked to ensure a logical flow of information. A thorough proofreading must be conducted to catch typographical errors, missing words, or grammatical mistakes that might have been overlooked. The language of the article must be improved.

Author Response

Dear Reviewer#

Thank you very much for spending the time to review our manuscript. Please find the detailed responses below and the corresponding revisions/corrections highlighted/in track changes in the re-submitted files.

The introduction provides a comprehensive overview of the importance of sugar beet in Egypt, the threats posed by Fusarium species to sugar beet cultivation, and the potential of Streptomyces isolates as biocontrol agents. It ss well-structured and informative, with a strong emphasis on the significance of the research.

Matherial and methods:

The methodology is strong in its overall structure and detail but could benefit from additional explanations in certain areas and improved clarity.

Thank you so much for your evaluations and valuable remarks on our manuscript.

Comments 1: Line 88: Clarify Sampling Locations and Number of Samples

Response 1: Thank you for pointing this out. We agree with this comment. Therefore, we have mentioned sampling location and number of samples from each location

Comments 2: Line 102: The text mentions incubating plates at 30°C for 15 days. It would be beneficial to briefly explain why this temperature and duration were chosen, linking it to the optimal conditions for the growth of actinobacteria.

Response 2: Need fully done

Comments 3: Line 105: The description of long-term storage in 15% glycerol at -80°C is clear. However, mentioning the rationale for choosing glycerol as the cryoprotectant and the typical duration for which cultures can be stored would be informative.
Response 3: Need fully done. Glycerol is used as a cryoprotectant as it disrupts the hydrogen bonding between water molecules, hence preventing the formation of ice crystals during freezing. This protects actinobacterial cells from ice crystal damage, and maintains their viability for future reculture.

Comments 4: Line 108: Citation?
Response 4: Need fully done

Comments 5: Line 110: The number identifying the strain?
Response 5: Need fully done

Comments 6: Line 111 and further: PDA provider?
Response 6: I cannot understand what is required

Comments 7: Line 166: name of the authors of the methodology cited
Response 7: Need fully done

Comments 8: Line 174: Clarify the specific purpose of using commercial fungicide Metalaxyl M + fludixonil (1 cm/kg) as the positive control. Explain why this particular fungicide was chosen.
Response 8: Need fully done

Comments 9: Line 184: Briefly explain the significance of these specific time points in relation to sugar beet growth stages.

Response 9: Need fully done

Comments 10: Table 2 Can those results be quantified to compare the actual quantity of produced enzymes?

Response 10: Actually, most of these enzymes were measured qualitatively

Comments 11: Table 5-6: statistical analysis is needed

Response 11: Statistical analysis were included in Tables 5-6 and LSD were calculated for the Treatments and varieties and for the interaction between them.

The attached file contains all comments, corrections, and answers to reviewers.

Round 2

Reviewer 1 Report

Comments and Suggestions for Authors

 Several points were not dissolved. For example, Metalaxyl M has no inhibitory activity against fungi, it is a chemical against Oomycetes .

Author Response

Dear Reviewer

Thanks a lot for your response

All points and comments for your review have been fully done and for the activity of Metalaxyl-M against fungi and especially Fusarium we sent some references which indicated this point 

Reviewer 3 Report

Comments and Suggestions for Authors

The article has been substantially enhanced in accordance with the reviewer's recommendations. It now comprehensively addresses the key issues and effectively engages the intended audience. I highly commend the authors' efforts in addressing the feedback and recommend that the article be accepted for publication in its current form.

Author Response

Thank you very much for your response and accepting our article Community Verified icon